# Unpacking antigypsism and support for solidarity-based actions: Implications of social class, discrimination awareness, group efficacy and emotions

Cristina Carmona-López[1,2]*, Ana Urbiola[3,2], Marisol Navas[1,2], Soledad de Lemus[3]

1 Department of Psychology, University of Almeria, Almería, Spain, 2 Centre of Study of Migration and Intercultural Relations (CEMyRI), University of Almeria, Almería, Spain, 3 Department of Social Psychology, University of Granada, Granada, Spain

* ccl319@inlumine.ual.es

## Abstract

Roma people, being the largest ethnic minority in Europe, continue to experience prejudice and structural discrimination. Moreover, there is low participation in Roma collective action and allies' solidarity-based actions. This pre-registered experimental research examines on samples from non-Roma population in Spain how social class, discrimination awareness, and group efficacy predict prejudice towards Roma and non-Roma participation in solidarity-based actions as allies in addition to test the role of intergroup emotions on those effects. In Study 1 (N = 870) social class and discrimination awareness were manipulated. Results showed that individuals assigned to the low social class condition exhibited more prejudice towards Roma in terms of stereotypes, emotions and discriminatory behaviors. Moreover, discrimination awareness condition indirectly predicted more participation in pro-Roma solidarity-based actions through increased outrage about about the situation of Roma. Study 2 (N = 1,000) confirmed the effect of social class on prejudice. Further, it showed two different paths for predicting solidarity-based actions: discrimination awareness (high vs. low) predicted higher participation in solidarity-based actions indirectly via empathy towards Roma people and outrage towards about the situation of Roma, whereas group efficacy (high vs. low) predicted participation in solidarity-based actions through hope in relation to the situation of Roma and empathy towards Roma. This experimental research highlights the need to address the role of Roma social class as a crucial factor in understanding prejudice and confirms the discrimination (via outrage and empathy) and efficacy (via hope) routes for promoting solidarity-based actions participation to support Roma rights and promote social equity.

**Data availability statement:** All data underlying the findings of this project are public and available on the Open Science Framework (OSF) at this link: https://osf.io/e3yzw. Additionally, the preregistrations for both studies are also available on OSF at the following links: https://osf.io/e8a4h (Study 1) and https://osf.io/dxnmq (Study 2).

**Funding:** This study was supported by the Rights, Equality and Citizenship (REC) Programme (2014–2020) of the European Union in the form of a research project grant awarded to AU (963122-ENGAGE-REC-AG-2020/REC-RDIS-DISC-AG-2020); by the Spanish Ministry of Science, Innovation and Universities (MCIU) in the form of a grant awarded to AU and SL (PID2022.141182NB.I00) and in the form of a salary for CC; and by the Spanish Ministry of Science, Innovation and Universities and the Agencia Estatal de Investigación (MCIU/AEI/10.13039/501100011033), co-funded by FSE+, in the form of a fellowship/salary for AU (RYC2022-035896-I). MN participated in the project but did not receive salary from these funders. The specific roles of these authors are articulated in the 'Author Contributions' section. The funders had no role in study design, data collection and analysis, decision to publish, or preparation of the manuscript.

**Competing interests:** The authors have declared that no competing interests exist.

## Introduction

Despite being the main ethnic minority in Europe (approximately 6 million [1]), Roma people still face *antigypsism* –negative attitudes or stereotypes attributed to Roma, whether intentional or unintentional [2]–. In 2023, 65% of European citizens surveyed acknowledged that there is widespread discrimination towards Roma in their country, 42% believed that being Roma is a disadvantage in hiring, 26% would feel uncomfortable about their child having Roma classmates at school, and 34% believed that their country's efforts to integrate Roma were ineffective [3]. However, at the European level, data on antigypsism and the historical context show substantial variation between different countries [4,5]. In Spain, the Roma population constitutes between 1% and 2.1% of the country's population, and they continue to face discrimination in different areas, such as education, employment (69% are unemployed) and housing [4]). In other countries as Hungary (7.05% of the country population), Slovakia (9.17% of the country population) or Rumania (8.32% of the country population) [6], high levels of discrimination are observed (e.g., 32% of the Roma population in Hungary and 45% in Romania are in a situation of severe material deprivation) [4].

In these countries many Roma children face discrimination at school: they are sent to "special" or Roma-only classes and schools, with fewer resources and poorer prospects, which means they have far fewer opportunities throughout their lives [4,7,8].During the Second World War, the Nazis murdered hundreds of thousands of Roma people, considering them to be "racially inferior" [9]. Later, between 1970 and 1990, approximately 90,000 Roma women were sterilized against their will in Czechoslovakia (now the Czech Republic and Slovakia) [10]. Spain presents worrying data regarding discrimination against Roma people, but it tends to present more encouraging data than the average for European countries [4]. Although its history has not been free from discrimination and persecution of the Roma population. One of the most recent events was during the dictatorship that the country experienced in the mid-20th century, when laws were used to criminalize the Roma way of life and their culture (including their language) was repressed, censored and turned into stereotypical folklore [11], with consequences that still determine the situation of the Roma today.

Traditionally, social psychology has focused on understanding the factors that contribute to such prejudice and discrimination towards the Roma population focusing on their ethnicity [12]. However, recent research suggests that social class may play a significant role in shaping these attitudes (e.g., [13,14]). This research aims to address this gap by adopting an intersectional approach in the study of interethnic relations, focusing on the role of social class attributed to Roma minority.

In addition to this hostile and unequal situation, social mobilization concerning the Roma community rights is scarce (e.g., [15,16]). Currently, Roma fight against antigypsism primarily takes place on digital platforms, and there is less activity in public spaces [17]. This limited social mobilization in support to Roma may be influenced by factors such as discrimination awareness [18], and belief in the efficacy of non-Roma individuals as allies to make a real impact in advocating for the rights of those who face discrimination daily. Importantly, in this research we study the influence of

discrimination awareness and group efficacy from the perspective of the advantaged group as allies –a perspective that has been underexplored from the majority viewpoint in prior literature (see [12]).

## The role of social class in antigypsism

People of lower social class are considered one of the social groups that face the most prejudice against [13]. In line with Wright [19], we understand *low social class* as a social position of people who do not have the educational or cultural resources to live above the poverty line. Adopting an intersectional approach to the study of prejudice towards ethnic minorities is not only necessary to capture the complexity of social inequalities but also has the potential to uncover how social class – often overlooked in traditional studies – intersects with other social categories to produce unique forms of disadvantage and discrimination. Previous research confirmed the role of classism in antigypsism, where participants showed more negative attitudes towards Roma from a lower social class than from high social class profile [14]. In line with the impact of intersectionality [20], their results demonstrated that stereotypes, intergroup emotions, and discrimination towards Roma are better predicted when both ethnicity and social class are considered interactively.

These findings align with previous research on the intersection of ethnicity and social class in shaping prejudice in the context of other ethnic minorities, such as Blacks in the United States (e.g., [21,22]). Other authors asserted that the discrimination faced by ethnic minorities may be *mostly* due to attribution to a lower social class [21]. Another, more equidistant perspective pointed out that both variables (ethnicity x social class) are simultaneously determining factors of prejudice and discriminatory attitudes (e.g., [14,23–25]). Building upon this, our research aims to further examine and confirm the role of social class in antigypsism. Specifically, we manipulated social class in order to test its influence on prejudice towards Roma and its potential interactive effects with discrimination awareness when predicting solidarity-based pro-Roma actions.

## What moves the advantaged groups to engage solidarity-based actions?

Among powerful factors involved in collective action participation, awareness of injustice/discrimination towards disadvantaged groups and group efficacy have been widely studied [26–30]. However, these factors have been studied mainly among disadvantaged groups' perspectives and less attention has been paid to how these mechanisms operate in the case of advantaged groups acting in favor of disadvantaged groups (cf. [31]).

Meta-analytic evidence has shown that identity, discrimination awareness and group efficacy have positive relations with solidarity-based actions intentions [31]. Being *aware of the discrimination* –or injustices– faced by minority or disadvantaged groups and acting accordingly is not that common, as the advantaged groups will try to maintain the *statu quo*. Alternatively, some advantaged group members can be aware of discrimination and try to do something to confront it. Szóstakowski and Besta [32] showed that the stronger perception and feeling of unjust treatment and identification with the discriminated groups were connected to the motivation to participate in solidarity-based actions in their favor. In the Roma context, perceiving social discrimination of this minority may be related to higher intentions to empower them –acquiring greater control in accessing opportunities [33]– as well as participate in solidarity-based actions [34]. Sometimes majority's awareness of discrimination towards a disadvantaged group mobilizes in its favor when it is related to an emotion (e.g., [30,35]), such as anger-outrage, empathy [36,37] or even pity in some contexts [18].

*Group efficacy* has been studied as a variable that plays a key role in explaining the tendency to participate in collective action [38–42], based on the idea that people will participate in collective action if they believe that their goals will be achieved [43,44]. Previous research found that both group efficacy and anger towards the minority situation felt by the majority group were positively related to the intention to engage in solidarity-based actions [45]. In the Roma context, Urbiola et al. [46] stated that group efficacy was related to the intention to participate in solidarity-based actions in favor of Roma. Also, the emotion of hope played an important role in the relationship between the perception of group efficacy and participation in collective actions in previous studies [47,48]. However, the main challenge for understanding the role of efficacy in social change is manipulating it, since the

obtained results with this goal are inconsistent [39,46]. In this research we experimentally manipulate discrimination awareness of Roma and group efficacy of non-Roma as potential predictors of solidarity-based actions for Roma rights engagement.

## Overview

We conducted two experimental studies from the non-Roma perspective with three main objectives. We aimed: (a) to confirm the role of social class in prejudice and discrimination against Roma people, (b) to analyze the effects of social class, awareness of discrimination towards Roma, and group efficacy as non-Roma allies on participation in solidarity-based actions with innovative experimental designs, and (c) to explore the role of emotions towards Roma people and the situation of Roma people as mediators in predicting solidarity-based actions.

This research seeks to bridge these gaps in literature by examining the role of social class in shaping prejudice towards Roma people and manipulating discrimination awareness of Roma and group efficacy as well as exploring emotions towards Roma people (pity and empathy) and towards the situation of Roma (outrage and hope) as influential factors in non-Roma participation in solidarity-based actions. Moreover, this research introduces an innovative experimental design by employing video-based manipulations, which not only strengthen the ecological validity but also offer valuable insights for future applied interventions. In the first study, we orthogonally manipulated social class and awareness of discrimination experienced by a Roma woman in the labor market using videos created *ad-hoc*, and we assessed their impact on prejudice and in non-Roma participation on solidarity-based actions as allies. In the second study, we extended this approach by also manipulating the group efficacy of non-Roma as allies.

## Study 1

This study employs a 2x2 design to examine how social class attributed to Roma people (low vs. high) affect prejudice towards Roma people but also analyze how awareness of discrimination towards Roma (low vs. high) affects intentions and participation in solidarity-based actions for Roma rights. Furthermore, we analyzed the mediating role of emotions towards Roma people and their situation in the relationship between awareness of discrimination towards Roma and participation in pro-Roma solidarity-based actions. Based on previous literature (e.g., [14,25,30,49]), the following hypotheses were pre-registered at Open Science Framework (OSF) (https://osf.io/e8a4h):

H1) A main effect of social class on prejudice, where participants assigned to the low social class condition will show more negative attitudes (will attribute fewer positive characteristics, more negative emotions, fewer positive emotions, and recommend less for a job) towards Roma target than participants assigned to the high social class condition.

H2) Participants assigned to the high awareness of discrimination condition will participate to a greater extent in solidarity-based actions in support of Roma people than those assigned to the low awareness of discrimination condition.

H3) An interaction effect between awareness of discrimination and social class is expected in the tendency to participate in solidarity-based actions, where participants who were assigned to the condition of high awareness of discrimination and low social class would show greater intention and participation in solidarity-based actions than the others.

H4) Empathy towards Roma people (H4a) and outrage towards the situation of Roma people (H4a) will mediate the effect of awareness of discrimination on solidarity-based actions.

H5) Participants with a greater perception of group efficacy of non-Roma people as allies will have higher scores in solidarity-based actions.

## Method

### Participants

Initially, it was preregistered that at least 400 participants (100 per experimental condition) would be included in the analysis. We hired a company specialized in sample collection for scientific research (Netquest). We require a cohort of 800 valid participants, selected to mirror a representative Spanish sample among their panelists based on gender, age, region

of origin, and social class. A sample of 1,101 participants was collected in January 2023. Of these, 227 were eliminated due to pre-established exclusion criteria: being a Roma person (23 participants) and not passing the attention check (204 participants). Additionally, four participants were eliminated for not having watched the video used as a manipulation. Based on Superpower [50] results showed that for four groups, a standard deviation of 1, and a final sample of 870 participants ($M_{age}$ = 44.42; $SD$ = 13.66), the statistical power is greater than a 95% for the pre-stablished analyses.

The final sample was composed by 49.5% participants that identified themselves as female, 49.4% as male and 0.7% identified themselves as non-binary. Regarding work, 64.4% reported being actively working, while 16% were unemployed, 7.1% were students, and 9.4% were retired. About participants' educational level, 0.5% did not end elementary studies, 3.9% finished their elementary studies, 10.7% graduated from secondary school, 37.5% did vocational training or baccalaureate, 31.8% had a university degree, 13.1% a master's degree and 2.5% had a Ph.D. Regarding the social class of the participants, 27.4% were in a high social class, 12.9% in an upper-middle social class, 26.5% in a medium social class, 12.5% in a lower-middle social class and 20.6% were in a low social class.

### Design and procedure

This is a 2x2 experimental study, where the social class of a Roma woman (low vs. high) and awareness of discrimination (low vs. high) are manipulated using a video created *ad-hoc*, in which a Roma woman was interviewed. In order to manipulate social class, we changed the neighborhood where she lived (city center vs. outskirts), the type of house (flat vs. social house), the family economy (they live quite well and didn't have economic problems vs. they faced economic problems every month), the type of work (lawyer vs. hairdresser), the appearance and clothes, and her accent and way of speaking (no-accent vs. an accent typically associated to people from a low socioeconomic status). In the case of high awareness of discrimination condition, the Roma woman said that she was not hired because of her ethnicity and provided real statistical data on the discrimination faced by Roma people in Spain. In contrast, in the low awareness of discrimination condition, the Roma woman explained that she was hired, and her colleagues were very kind, and provided neutral socio-demographic data about Roma people in Spain (e.g., they compose the 1.5% of Spanish population, Roma people are in several European countries). See https://osf.io/e3yzw at Open Science Framework for detailed materials, such as questionaries, databases, and the scripts of the manipulations.

After watching the video, an online questionnaire was completed on Qualtrics and lasted approximately 12 minutes. At the beginning of the questionnaire, participants were asked about their consent and informed about the anonymity and confidentiality of their answers, as well as the voluntary nature of their participation. Informed consent was obtained in written electronic form: participants could only access the questionnaire after actively indicating their agreement by clicking an "I agree to participate" button. No minors took part in this study; therefore, no parental or guardian consent was required. The study was approved by the Human Research Bioethics Committee at the researchers' university.

### Instruments

*Manipulation Checks.* Three manipulation checks were performed to confirm that the participants had understood the manipulation. First, they were asked to explain in a few words the content of the interview they just have seen. After that, they were asked for their awareness of discrimination suffered by the target (a young Roma woman called Carmen) with a 5-point scale (1 = *Nothing*; 5 = *Very much*) with following question: *According to what Carmen says, to what extent do you think she has been discriminated for being a Roma person?* Secondly, participants were asked to place Carmen in the social class they thought she belonged to on a 5-point scale (1 = *low social class*; 5 = *high social class*). Finally, participants were asked to select the range in which they thought Carmen's monthly income was (1 = *Less than €500*; 5 = *More than €2000*). For clarification and parsimony of the presentation of results, social class condition will be presented as 'high', however, it is really an 'upper-middle' social class, since the percentage of Roma in Spain that belong to a high social class is extremely low.

*Stereotypes towards People like the Target.* A 9-items scale was used [51] to evaluate three stereotypes' dimensions (morality, sociability, and competence). Participants indicated to what extent they thought that people like Carmen were: honest, sincere, and trustworthy (morality dimension; α = .94), pleasant, friendly, and warm (sociability dimension; α = .91), intelligent, skillful and competent (competence dimension; α = .85), using a 5-point scale (1 = *Nothing*; 5 = *Very much*). High scores indicated more positive characteristics attributed to those people.

*Emotions towards People like the Target.* This variable was measured with the emotion's subscale of the Prejudice Attitude Test (PAT, [52]), composed by seven items. Participants had to indicate to what extend they felt each emotion towards Roma people like Carmen, using a 5-points scale (1 = *Nothing*, 5 = *Very much*). Three emotions were positive (admiration, sympathy, and respect, α = .85) and four negatives (mistrust, discomfort, insecurity, and indifference, α = .90). Higher scores indicated more intense emotions (positive or negatives) towards the people like the target.

*Recommendation for a Job Position.* This variable was measured by an item. Participants were asked to indicate to what extend they would recommend people like Carmen for a work, using a 7-points scale (1 = *I would not recommend people like her at all*; 7 = *I would totally recommend people like her*). High scores in the scale indicated high degree of recommendation for a job position.

*Emotions towards Roma People.* This variable was measured with two items based on Lantos et al. [18]. Participants were asked to indicate at what extent they feet pity or empathy towards Roma people, using a 5-point scale (1 = *Nothing*; 5 = *Very much*).

*Emotions towards the Situation of Roma People.* This variable was measured with three items. Participants were asked to indicate to what extend they felt outrage-anger ($r = .65$), and hope towards the situation of Roma people in Spain, using a 5-points scale (1 = *Nothing*; 5 = *Very much*).

*Group Efficacy.* This variable was measured with a 4-items scale, based on van Zomeren et al. [53]. Participants were asked to indicate their level of agreement with the effectiveness of non-Roma as allies contributing for the social change (e.g., change the situation of inequality of the Roma), using a 7-point Likert scale (1 = *Totally disagree*; 7 = *Totally agree*). Higher scores indicated higher perceptions of group efficacy (α = .93).

*Pro-Roma Solidarity-Based Actions Intentions.* This variable was measured with eight items based on collective action intentions' scale [54,55]. Participants had to indicate to what extent they were willing to participate in different types of solidarity-based actions (e.g., write claims to public institutions, attend protests), using a 5-point scale (1 = *Nothing*; 5 = *Very much*). Higher scores showed greater intention (or willingness) to participate in actions supporting Roma equality (α = .94).

*Pro-Roma Solidarity-Based Actions Participation (Sign a Petition).* Participants were asked to read a petition defending higher equality of the Roma in educational contexts and had to choose between signing it or not.

The order of the presentation of solidarity-based actions variables (intentions and participation) was counterbalanced.

## Analytic strategy

To examine effects on prejudice towards Roma, we estimated a between-subjects multivariate general linear model (MANOVA) with the two experimental factors entered as independent variables (Social Class: low vs. high; Discrimination Awareness: low vs. high).

We tested mediation models entering simultaneously as independent variables (X): Social Class Condition (low = 0; high = 1) and Discrimination Awareness Condition (low = 0; high = 1). The mediators (M) were the emotion variables (outrage, empathy, hope and pity), and the outcomes (Y) were solidarity-based actions intentions and participation (petition signing). We report unstandardized estimates (B), Standard Error (SE) and 95% CIs. Indirect effects are based on 10,000 bootstrapping; significance is inferred when the 95% CI excludes zero.

 

## Results and discussion

Table 1 shows the descriptive statistics and the bivariate correlations of the studied variables. To ensure that the manipulation have worked as expected, we performed two *t* tests for independent samples. The results showed that participants assigned to the low social class condition ($M = 1.50$, $SD = 0.62$) and those assigned to the high social class condition ($M = 3.17$, $SD = 0.82$), scored significantly different on the social class manipulation check ($t_{(868)} = -34.10$, $p < 0.001$, $d = 2.29$). Similarly, participants assigned to the low discrimination awareness condition ($M = 1.66$, $SD = 0.94$) and those assigned to the high discrimination awareness condition ($M = 4.28$, $SD = 0.89$), scored significantly different on the manipulation check for discrimination awareness ($t_{(868)} = -40.51$, $p < 0.001$, $d = 2.86$).

### Effects of social class on prejudice towards Roma people

A multivariate analysis was carried out including the two independent variables. Significant effects of social class were found on all the dependent variables of prejudice (morality, $F = 43.06$, $p < 0.001$, $\eta^2_p = .05$; sociability $F = 21.07$, $p < 0.001$, $\eta^2_p = .02$; and competence stereotypes, $F = 43.62$, $p < 0.001$, $\eta^2_p = .05$; positive emotions, $F = 38.80$, $p < 0.001$, $\eta^2_p = 0.04$; negative emotions, $F = 38.51$, $p < 0.001$, $\eta^2_p = .04$; and recommendation for a job, $F = 44.38$, $p < 0.001$, $\eta^2_p = .05$). As expected, participants assigned to the experimental condition of low social class (vs. high social class) showed more prejudice towards Roma people like the target presented in the video (see Fig 1).

### Effects on solidarity-based actions

Univariate analyses were conducted, revealing no significant effects of discrimination awareness on solidarity-based actions intentions ($F_{(1,870)} = 0.02$, $p = 0.888$) or its interaction with social class ($F_{(1,870)} = 0.12$, $p = 0.729$). Binary logistic regression analysis was conducted to investigate the effect of discrimination awareness (coding: low discrimination awareness = 0, and high discrimination awareness = 1) and social class (coding: low social class = 0, high social class = 1) on solidarity-based actions participation. The model was not significant ($\chi^2_{(1)} = 1.81$, $p = 0.405$), with no significant difference between conditions ($p = 0.378$) or interaction effects of discrimination awareness and social class on solidarity-based actions participation ($p = 0.645$).

**Table 1. Descriptive statistics and bivariate correlations of Study 1.**

| | M | SD | 1 | 2 | 3 | 4 | 5 | 6 | 7 | 8 | 9 | 10 | 11 | 12 | 13 |
|---|---|---|---|---|---|---|---|---|---|---|---|---|---|---|---|
| 1. Stereotypes of morality | 3.50 | 0.98 | – | | | | | | | | | | | | |
| 2. Stereotypes of sociability | 3.74 | 0.84 | .81** | – | | | | | | | | | | | |
| 3. Stereotypes of competence | 3.89 | 0.73 | .74** | .77** | – | | | | | | | | | | |
| 4. Positive emotions | 3.62 | 0.86 | .73** | .71** | .68** | – | | | | | | | | | |
| 5. Negative emotions | 2.13 | 0.93 | −.70** | −.65** | −.58** | −.73** | – | | | | | | | | |
| 6. Recommendation for a job | 3.92 | 1.05 | .66** | .62** | .59** | .71** | −.67** | – | | | | | | | |
| 7. Group efficacy | 4.73 | 1.47 | .57** | .54** | .48** | .58** | −.50** | .53** | – | | | | | | |
| 8. Pity towards Roma people | 1.96 | 0.97 | .14** | .12** | .09* | .14** | −.01 | .10* | .18** | – | | | | | |
| 9. Empathy towards Roma people | 3.03 | 1.09 | .52** | .50** | .44** | .56** | −.47** | .49** | .56** | .25** | – | | | | |
| 10. Outrage towards the situation of Roma | 2.52 | 1.00 | .36** | .37** | .30** | .37** | −.24** | .33** | .39** | .43** | .47** | – | | | |
| 11. Hope towards the situation of Roma | 2.92 | 1.07 | .44** | .42** | .38** | .50** | −.40** | .43** | .53** | .24** | .55** | .39** | – | | |
| 12. Solidarity-based actions intentions | 2.59 | 1.00 | .51** | .47** | .42** | .54** | −.46** | .51** | .60** | .25** | .58** | .52** | .50** | – | |
| 13. Solidarity-based actions participation (petition) | – | – | .29** | .28** | .24** | .31** | −.28** | .30** | .35** | .08* | .33** | .26** | .30** | .52** | – |

* indicates $p < 0.05$.

** indicates $p < 0.001$.

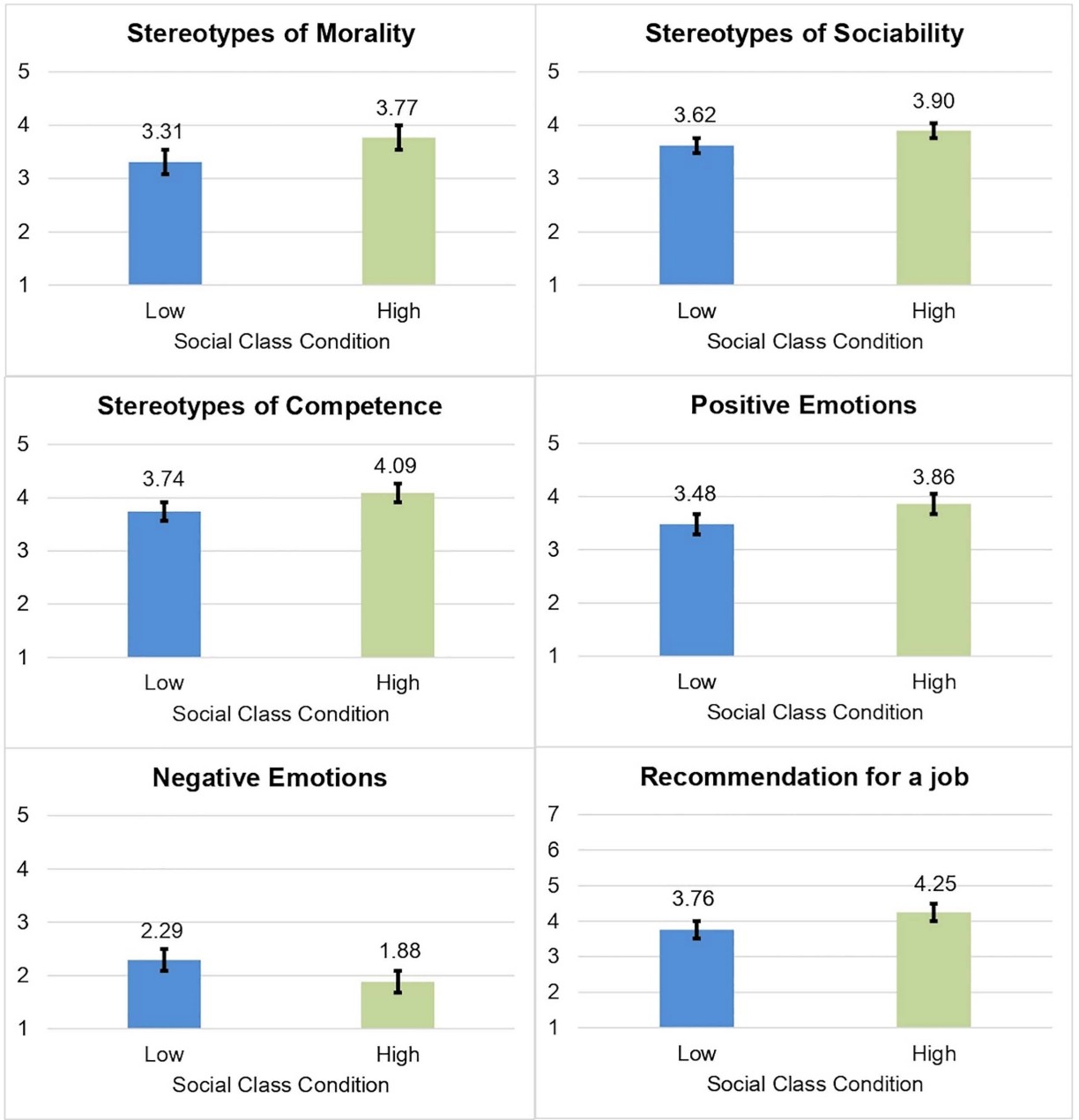

**Fig 1. Effects of social class on prejudice.** Error bars show 95% CIs. Stereotypes refer to the attribution of such a quality to people with the target characteristics.

However, when mediation analyses with emotions were performed with 10,000 bootstrapping samples using GLM Mediation Model on Jamovi, indirect effects were found. We carried out a model with the two main independent variables [discrimination awareness ($X_1$) and social class ($X_2$)] as predictors, the emotions towards the situation of Roma [outrage ($M_1$) and hope ($M_2$)] and towards Roma people [pity ($M_3$) and empathy ($M_4$)] as mediators, and solidarity-based actions on intentions ($Y_1$) and participation (dichotomous) ($Y_2$) as dependent variables (see Fig 2 and Table 2). Results showed

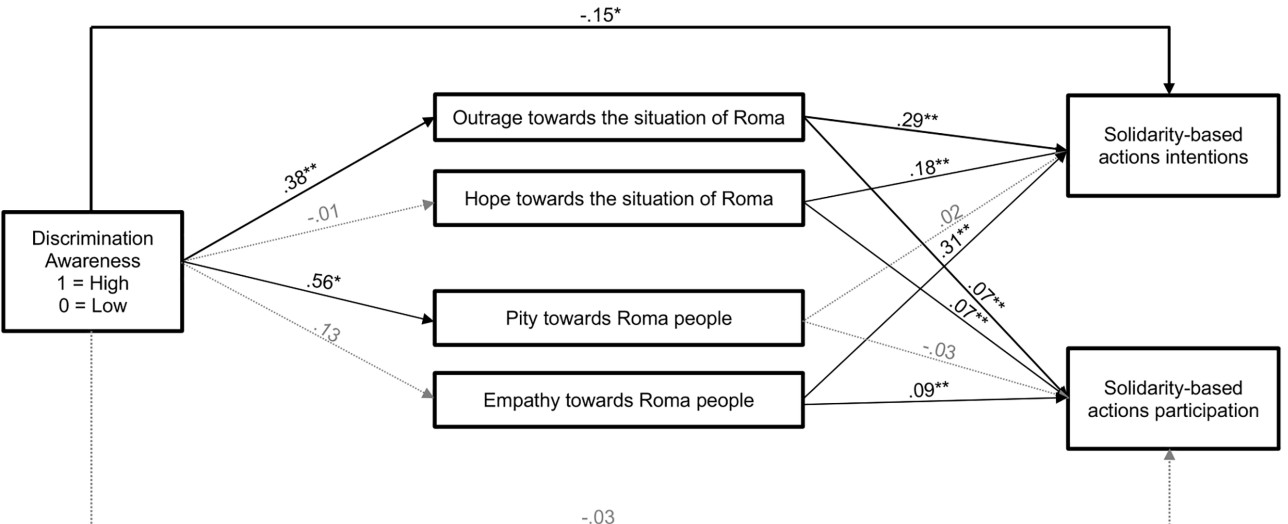

**Fig 2. Mediational paths of pity and empathy towards Roma people and outrage and hope towards the situation of Roma in the relation of discrimination awareness and solidarity-based actions.** * indicates $p < 0.05$. ** indicates $p < 0.001$. Model includes Social Class and Discrimination Awareness simultaneously as predictors on all paths (a, b, c'). However, the paths for Social Class Condition are not shown to clarify the figure due to not significant effects.

**Table 2. Direct and indirect effects through emotions of discrimination awareness towards Roma and social class and intentions and participation of non-Roma in solidarity-based actions.**

|  | Solidarity-Based Actions Intentions | | Solidarity-Based Actions Participation | |
| --- | --- | --- | --- | --- |
|  | Coeff. | 95% C. I. | Coeff. | 95% C. I. |
| Direct Effect of Discrimination Awareness | **−.146 (.057)** | **[-0.263, -0.034]** | −.054 (.035) | [-0.122, 0.014] |
| Direct Effect of Social Class | .035 (.053) | [-0.070, 0.141] | .010 (.032) | [-0.053, 0.072] |
| D.A. Indirect Effect via Outrage | **.110 (.024)** | **[0.066, 0.167]** | **.026 (.009)** | **[0.012, 0.047]** |
| D.A. Indirect Effect via Hope | −.002 (.014) | [-0.034, 0.026] | .001 (.005) | [-0.013, 0.010] |
| D.A. Indirect Effect via Pity | .011 (.017) | [-0.022, 0.047] | −.015 (.010) | [-0.036, 0.006] |
| D.A. Indirect Effect via Empathy | .039 (.025) | [-0.012, 0.091] | .012 (.008) | [-0.002, 0.029] |
| S.C. Indirect Effect via Outrage | .005 (.020) | [-0.036, 0.047] | .001 (.005) | [-0.009, 0.012] |
| S.C. Indirect Effect via Hope | .016 (.014) | [-0.010, 0.047] | .006 (.005) | [-0.003, 0.019] |
| S.C. Indirect Effect via Pity | −.001 (.002) | [-0.011, 0.002] | .001 (.002) | [-0.002, 0.009] |
| S.C. Indirect Effect via Empathy | .024 (.024) | [-0.023, 0.072] | .007 (.007) | [-0.006, 0.023] |
| Total D.A. Indirect Effect | .012 (.074) | [-0.130, 0.152] | −.032 (.036) | [-0.103, 0.038] |
| Total S.C. Indirect Effect | .079 (.071) | [-0.060, 0.221] | .026 (.035) | [-0.041, 0.096] |

Statistically significant coefficients are shown in bold. D.A. = Discrimination awareness; S.C. = Social class.

direct negative effects of discrimination awareness ($X_1$) on solidarity-based actions intentions ($b = −.15$, $SE = .06$, C.I. 95% [−0.263, −0.033], $z = −2.55$, $p = 0.011$), but no relationship of discrimination awareness on solidarity-based actions participation was found. Results also showed indirect positive effects of discrimination awareness through outrage ($M_1$) on both types of solidarity-based actions: intentions ($b = .11$, $SE = .02$, C.I. 95% [0.066, 0.167], $z = 4.55$, $p < 0.001$) and participation ($b = .03$, $SE = .01$, C.I. 95% [0.012, 0.047], $z = 3.01$, $p = 0.003$).

In relation to group efficacy a simple linear regression analysis showed that participants with higher group efficacy had higher scores in solidarity-based actions intentions ($R^2 = .36$; $b = .41$, $SE = .02$, $\beta = .60$, $t = 22.06$, $p < 0.001$). In the same line, logistic regression analysis was found significant on solidarity-based actions participation ($\chi^2_{(1)} = 119.86$, $p < 0.001$, $b = .60$, $SE = .06$, $Wald_{(1)} = 94.74$, Nagelkerke $R^2 = .18$).

Synthesizing, in line with Hypothesis 1 and with previous research (e.g., [15,16,25,49], there is a main effect of social class on prejudice, where participants assigned to the lower social class showed more prejudice towards Roma people. In relation to Hypotheses 2 and 3, no direct effect of discrimination awareness or its interaction with social class on intentions or participation in solidarity-based actions in favor of Roma was found. However, in line with Hypothesis 4, there is a relationship of discrimination awareness in solidarity-based actions through increased emotion of outrage towards the situation of Roma people [36,37]. With respect to this mediation analysis, it was found that discrimination awareness was directly related to a lower intention to participate in solidarity-based actions. These results may suggest the presence of a suppression effect [56] such that the total effect was not significant because the indirect effect via emotions was positive and may have masked a negative effect of discrimination awareness on solidarity-based actions intentions. Finally, in line with Hypothesis 5, group efficacy was found to be positively related to greater intention and participation in solidarity-based actions [31,40–43].

## Study 2

The present study aims to replicate and extend the findings obtained in Study 1, ensuring the robustness of the results and exploring in greater depth the factors that influence solidarity-based actions in favor of the Roma population. This second study incorporates the manipulation of group efficacy, given that previous results of its predictive power for solidarity-based actions.

This study was pre-registered on the Open Science Framework (OSF) platform (https://osf.io/dxnmq). Based on Study 1 and previous literature [16,25,30,36,37,42,44,47,49] the following hypotheses are established:

H1) A main effect of the social class condition on prejudice is expected. Participants assigned to the low social class condition will show more negative attitudes than participants assigned to the high social class condition.

H2) An indirect effect of discrimination awareness condition on solidarity-based actions is expected through empathy towards Roma people. Pity towards Roma people is not expected to mediate this relationship. Due to previous literature [30], the direct effect of discrimination awareness condition on solidarity-based actions (intentions and participation) will be explored.

H3) An indirect effect of discrimination awareness on solidarity-based actions is expected through the emotion of outrage toward the situation of Roma people.

H4) Based on the Study 1, we expect a direct effect of group efficacy on solidarity-based actions. Participants assigned to the high group efficacy condition will show higher tendency to participate in solidarity-based actions.

H5) An indirect effect of group efficacy on solidarity-based actions is expected through the emotion of empathy towards Roma people (H5.1), and through hope and outrage towards the situation of Roma (H5.2).

H6) An interaction effect is expected between discrimination awareness and group efficacy on solidarity-based actions. Participants assigned to the high discrimination awareness and high group efficacy conditions will participate more in solidarity-based actions (intentions and participation).

## Method

### Participants

Based on Superpower [50] it was preregistered that at least 640 participants would be included in the analyses. Results showed that for eight groups, a standard deviation of 1, and 80 participants in each group, the statistical power is greater than a 99% for the pre-stablished analyses. In addition, this simulation also gives the *t*-test for pairwise comparison, where the statistical power was found to be identical to ANOVA's results.

In line with Study 1, we hired a company specialized in sample collection for scientific research (Netquest). We require a cohort of 1,000 valid participants, carefully selected for a representative Spanish sample among their panelists. This entails adhering to pre-established exclusion criteria, ensuring alignment across gender, age, region of origin, and social class.

Finally, a sample of 1,103 participants was collected in April 2024, but 103 were eliminated due to pre-registered exclusion criteria: being Roma (24 participants) and not passing the attention check (where they had to mark the number 2 to prove that they were reading the questionnaire; 79 participants). The final number of participants was composed by 1,000 ($M_{age}$ = 48.82; $SD$ = 16.64) of which, 51.1% identified themselves as female, 48.4% as male and 0.3% identified themselves as non-binary. Regarding work, 56.7% reported being actively working, while 11% were unemployed, 6.4% were students, and 23.6% were retired. About participants' educational level, 0.5% did not end elementary studies, 4% finished elementary studies, 7.7% graduated from secondary school, 36.5% did vocational training or baccalaureate, 32.8% had a university degree, 15.6% a master's degree and 2.9% a Ph.D. Regarding the social class of the participants, 29.8% were in a high social class, 14.7% in an upper-middle social class, 30.5% in a medium social class, 12.0% in a lower-middle social class and 13% were in a low social class.

## Design and procedure

This is a 2x2x2 experimental study, where the social class of a Roma woman (low vs. high), discrimination awareness (low vs. high), and group efficacy of non-Roma as allies (low vs. high) are manipulated. Social class and discrimination awareness were manipulated as in the Study 1. The manipulation of group efficacy was carried out by an online newspaper article about the efficacy of non-Roma in bringing about social change in favor of Roma rights following the procedures of Urbiola et al. [46]. Participants randomly read one of two possible news items: high non-Roma allies' group efficacy or low group efficacy (see the materials in Open Science Framework platform at https://osf.io/e3yzw).

This study was conducted by an online questionnaire on the Qualtrics platform and took an average of 13 minutes. At the beginning of the questionnaire, participants were asked about their consent and informed about the anonymity and confidentiality of their answers, as well as the voluntary nature of their participation. As in the previous study, informed consent was obtained in written electronic form: participants could only access the questionnaire after actively indicating their agreement by clicking an "I agree to participate" button. No minors took part in this study; therefore, no parental or guardian consent was required. This study was approved by the Human Research Bioethics Committee at the researchers' university.

## Instruments

As in the Study 1, the following variables and instruments were included: stereotypes towards people like the target [morality (α = .93), sociability (α = .91), and competence (α = .88)], emotions towards people like the target [positive (α = .85) and negative (α = .89)], recommendations for a job position, emotions towards Roma people (pity and empathy), emotions towards the situation of Roma (outrage-anger [$r$ = .66] and hope), solidarity-based actions intentions (α = .94), and participation (dichotomous). The same manipulation checks from Study 1 were used for both social class and discrimination awareness.

**Group Efficacy's Manipulation Checks.** Participants were asked to explain in their own words the article they just read. Secondly, they rated the efficacy of non-Roma actions as allies to promote social change on a 5-point scale (1 = *Not at all effective*; 7 = *Very effective*).

## Analytic strategy

The analyses were performed in the same way as in Study 1. In this study, Group Efficacy (low = 0; high = 1) was also added as a predictor variable (X).

## Results and discussion

Table 3 shows the descriptive statistics and the bivariate correlations of the studied variables. To ensure the effectiveness of the manipulation, we performed three $t$ tests for independent samples. The results show that participants assigned to the low social class condition ($M = 1.62$, $SD = 0.67$) and those assigned to the high social class condition ($M = 3.11$, $SD = 0.79$), scored significantly different on the social class manipulation check as in Study 1 ($t_{(998)} = -32.39$, $p < 0.001$, $d = 2.03$). Similarly, participants assigned to the low discrimination awareness condition ($M = 1.64$, $SD = 0.88$) and those assigned to the high discrimination awareness condition ($M = 4.36$, $SD = 0.81$), scored significantly different on the manipulation check for discrimination awareness ($t_{(997)} = -50.97$, $p < 0.001$, $d = 3.22$). Finally, group efficacy was measured using the group efficacy scale. Results showed that participants assigned to the low group efficacy condition ($M = 2.57$, $SD = 1.67$) and those assigned to the high group efficacy condition ($M = 5.25$, $SD = 1.56$), scored significantly different on the manipulation check for group efficacy ($t_{(998)} = -26.20$, $p < 0.001$, $d = 1.66$).

### Effects of social class on prejudice towards Roma people

Significant multivariate effects of social class were found on the dependent variables of prejudice (morality, $F = 22.38$, $p < 0.001$, $\eta^2_p = .02$; sociability, $F = 13.53$, $p < 0.001$, $\eta^2_p = .01$; and competence stereotypes, $F = 29.82$, $p < 0.001$, $\eta^2_p = .03$; positive emotions, $F = 30.93$, $p < 0.001$, $\eta^2_p = .03$; negative emotions, $F = 13.93$, $p < 0.001$, $\eta^2_p = .01$; and recommendation for a job, $F = 63.12$, $p < 0.001$, $\eta^2_p = .06$). As expected, participants assigned to the low (vs. high) social class condition showed more prejudice towards Roma people (see Fig 3).

### Effects on solidarity-based actions

Univariate analysis' results did not show significant effects of social class ($F_{(1,1000)} = 0.27$, $p = 0.605$) or its interaction with discrimination awareness ($F_{(1,1000)} = 0.03$, $p = 0.859$) on solidarity-based actions intentions. Results neither showed significant effects of group efficacy condition on solidarity-based actions intentions ($F_{(1,1000)} = 1.54$, $p = 0.215$). In the same line, we performed logistic regression analysis with the three independent variables [social class, discrimination awareness, and group efficacy] (coding: low = 0; high = 1) on solidarity-based actions participation (dichotomous). Results showed no significant effects of the model ($\chi^2_{(3)} = 0.80$, $p = 0.849$).

**Table 3. Descriptive statistics and bivariate correlations of Study 2.**

| | M | SD | 1 | 2 | 3 | 4 | 5 | 6 | 7 | 8 | 9 | 10 | 11 | 12 |
|---|---|---|---|---|---|---|---|---|---|---|---|---|---|---|
| 1. Stereotypes of morality | 3.56 | 0.98 | – | | | | | | | | | | | |
| 2. Stereotypes of sociability | 3.79 | 0.86 | .84** | – | | | | | | | | | | |
| 3. Stereotypes of competence | 3.91 | 0.82 | .79** | .82** | – | | | | | | | | | |
| 4. Positive emotions | 3.73 | 0.86 | .69** | .70** | .67** | – | | | | | | | | |
| 5. Negative emotions | 2.00 | 0.89 | −.57** | −.52** | −.49** | −.64** | – | | | | | | | |
| 6. Recommendation for a job | 4.02 | 1.00 | .59** | .58** | .59** | .70** | −.59 ** | – | | | | | | |
| 7. Pity towards Roma people | 2.01 | 1.01 | .09* | .06 | .02 | .16** | .01 | .12** | – | | | | | |
| 8. Empathy towards Roma people | 3.12 | 1.03 | .46** | .46** | .39** | .54** | −.40** | .49** | .21** | – | | | | |
| 9. Outrage towards the situation of Roma | 2.51 | 1.00 | .32** | .28** | .23** | .38** | −.22 ** | .34** | .43** | .48** | – | | | |
| 10. Hope towards the situation of Roma | 3.02 | 1.09 | .44** | .43** | .36** | .53** | −.37** | .44** | .23** | .56** | .42** | – | | |
| 11. Solidarity-based actions intentions | 2.62 | 1.04 | .43** | .39** | .37** | .52** | −.35 ** | .47** | .23** | .55** | .52** | .51** | – | |
| 12. Solidarity-based actions participation (petition) | – | – | .28** | .26** | .26** | .32** | −.26 ** | .34** | .15** | .32** | .32** | .30** | .52** | – |

\* indicates $p < 0.05$.

\** indicates $p < 0.001$.

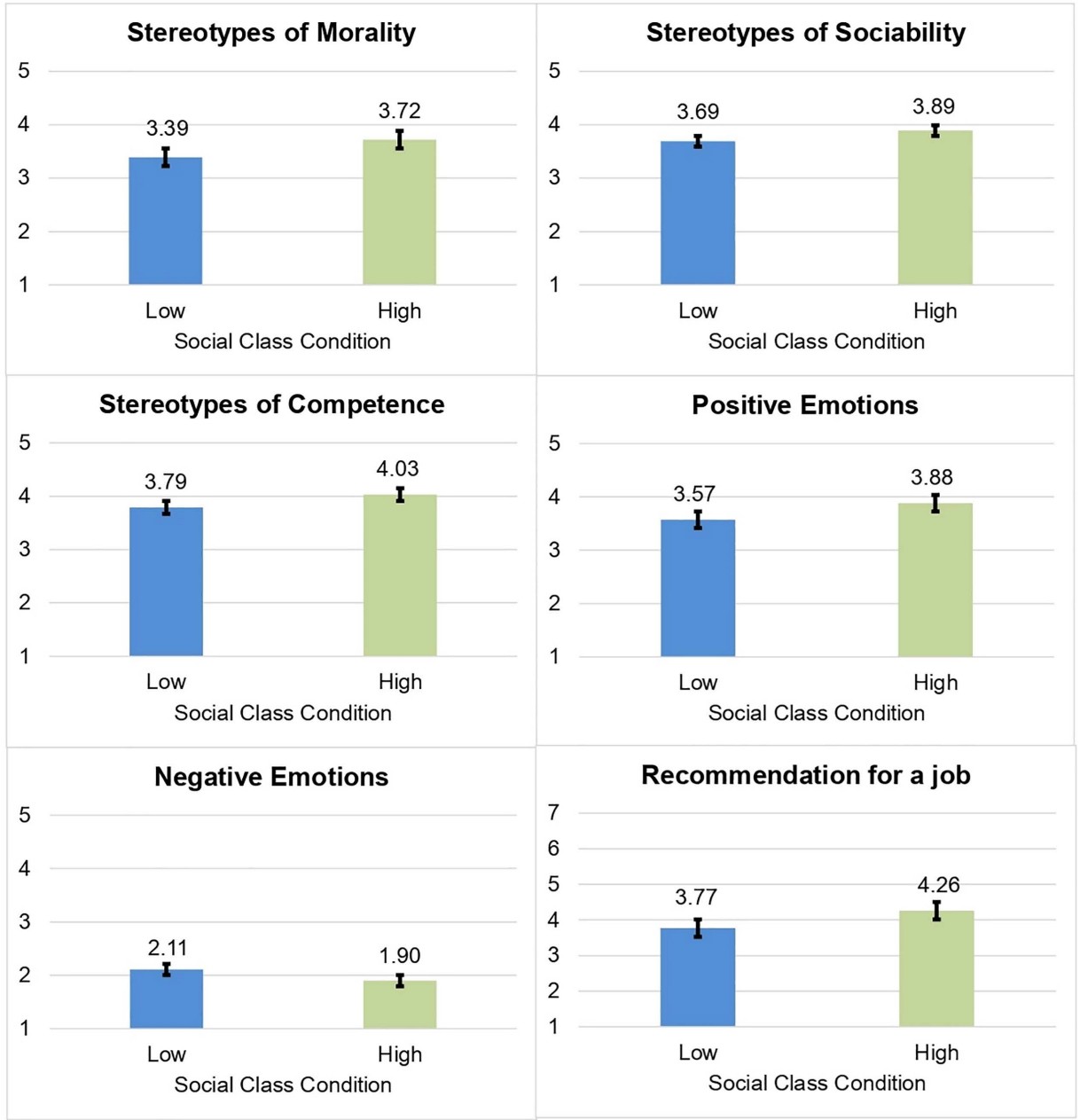

**Fig 3. Effects of social class on prejudice.** Error bars show 95% CIs. Stereotypes refer to the attribution of such a quality to people with the target characteristics.

The interaction of discrimination awareness and group efficacy did not have effects on solidarity-based actions intentions ($F_{(1, 1000)}$ = 0.10, $p$ = 0.756). However, we found a main effect of discrimination awareness on solidarity-based actions intentions ($F_{(1,1000)}$ = 6.45, $p$ = 0.011, $\eta^2_p$ = .006), where participants assigned to the high discrimination awareness condition had higher intention to participate in solidarity-based actions ($M_{low\ discrimination}$ = 2.53, $M_{high\ discrimination}$ = 2.70). No effect of discrimination awareness on solidarity-based actions participation was found ($X^2_{(1)}$ = 0.87, $p$ = 0.929), neither of its interaction with group efficacy condition ($p$ = 0.798).

Mediation analyses were performed using GLM Mediation Model on Jamovi, using 10,000 bootstrap samples. We examine whether outrage ($M_1$) and hope ($M_2$) towards the situation of Roma, and empathy ($M_2$) and pity ($M_3$) emotions towards Roma people will mediate the relationship between discrimination awareness ($X_1$), group efficacy ($X_2$), or social class ($X_3$) –entered simultaneously as predictors in all paths of the model– and solidarity-based actions intentions ($Y_1$) and participation (dichotomous) ($Y_2$) (see Fig 4 and Table 4).

Results showed no direct relationship between discrimination awareness ($X_1$) and solidarity-based actions: intentions ($b = -.03$, $SE = .05$, C.I. 95% [−0.133, 0.072], $z = -0.62$, $p = 0.536$) or participation ($b = -.05$, $SE = .03$, C.I. 95% [−0.108, 0.007], $z = -1.77$, $p = 0.077$). However, we found indirect effects of discrimination awareness on both types of solidarity-based actions through outrage towards the situation of Roma (intentions, $b = .13$, $SE = .02$, C.I. 95% [0.085, 0.182], $z = 5.54$, $p < 0.001$; participation, $b = .04$, $SE = .01$, C.I. 95% [0.024, 0.066], $z = 4.30$, $p < 0.001$) and empathy towards Roma people (intentions, $b = .06$, $SE = .02$, C.I. 95% [0.018, 0.099], $z = 2.81$, $p = 0.005$; participation, $b = .01$, $SE = .01$, C.I. 95% [0.004, 0.027], $z = 2.35$, $p = 0.019$).

Unexpectedly, the direct effect of group efficacy ($X_2$) on solidarity-based actions was non-significant for both intentions ($b = -.03$, $SE = .05$, C.I. 95% [−0.127, 0.073], $z = -0.59$, $p = 0.556$) or participation ($b = -.02$, $SE = .02$, C.I. 95% [−0.074, 0.039], $z = -0.62$, $p = 0.534$). However, the indirect effects through hope towards the situation of Roma (intentions, $b = .04$, $SE = .02$, C.I. 95% [0.010, 0.075], $z = 2.40$, $p = 0.016$; participation, $b = .01$, $SE = .01$, C.I. 95% [0.003, 0.023], $z = 2.07$, $p = 0.038$) and empathy towards Roma (intentions, $b = .05$, $SE = .02$, C.I. 95% [0.013, 0.095], $z = 2.59$, $p = 0.010$; participation, $b = .01$, $SE = .01$, C.I. 95% [0.004, 0.026], $z = 2.21$, $p = 0.026$) were significant.

Summarizing, the results showed in line with Hypothesis 1, a main effect of social class on prejudice towards Roma people, where participants assigned to the lower social class reported higher prejudice towards Roma [15,16,25,49]. However, no direct effects of group efficacy on solidarity-based actions or its interaction with awareness of discrimination

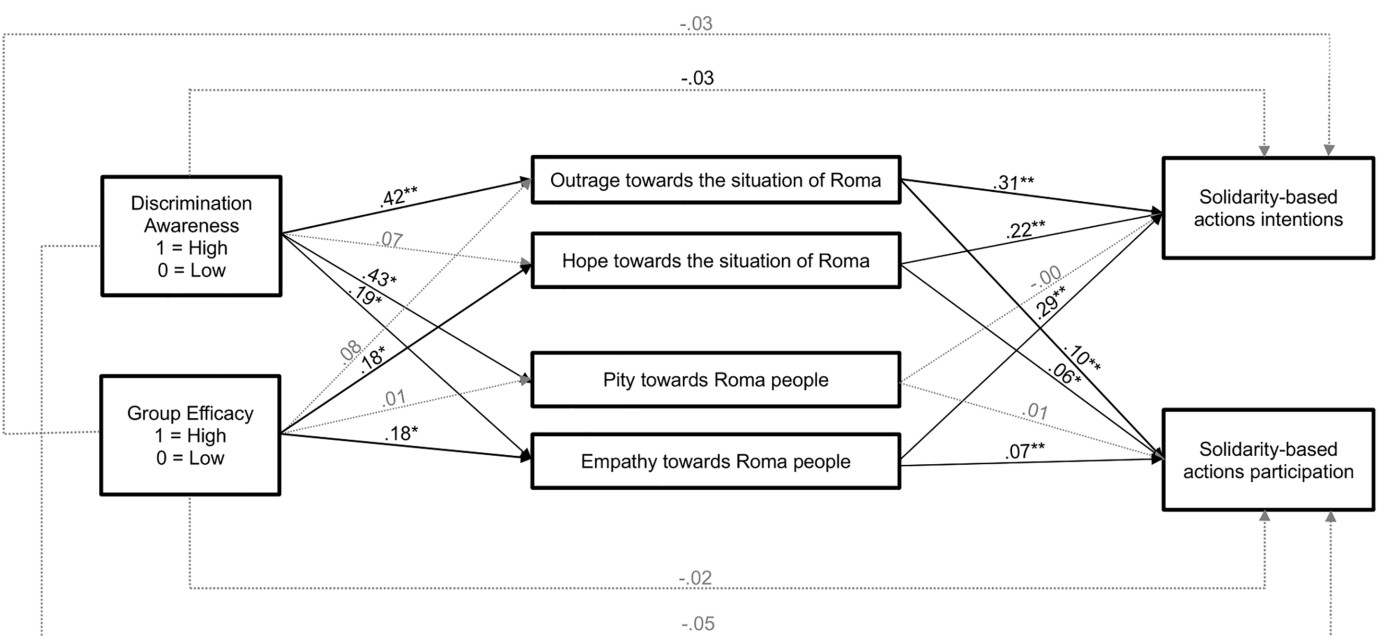

**Fig 4. Indirect effects through emotions.** * indicates $p < 0.05$. ** indicates $p < 0.001$. Model includes Social Class, Discrimination Awareness and Group Efficacy simultaneously as predictors on all paths (a, b, c'). However, the paths for Social Class Condition are not shown to clarify the figure due to not significant effects.

**Table 4. Direct and indirect effects of discrimination awareness, group efficacy, and social class and the intentions and participation of non-Roma in solidarity-based actions in Study 2.**

| | Solidarity-Based Actions Intentions | | Solidarity-Based Actions Participation | |
|---|---|---|---|---|
| | Coeff. | 95% C. I. | Coeff. | 95% C. I. |
| Direct Effect of Discrimination Awareness | −.032 (.052) | [-0.133, 0.072] | −.052 (.029) | [-0.108, 0.007] |
| Direct Effect of Group Efficacy | −.030 (.051) | [-0.127, 0.073] | −.018 (.029) | [-0.074, 0.039] |
| Direct Effect of Social Class | .002 (.050) | [-0.097, 0.102] | −.013 (.028) | [-0.071, 0.041] |
| D.A. Indirect Effect via Outrage | **.128 (.023)** | **[0.085, 0.182]** | **.042 (.010)** | **[0.024, 0.066]** |
| D.A. Indirect Effect via Hope | .016 (.015) | [-0.012, 0.049] | .004 (.004) | [-0.003, 0.015] |
| D.A. Indirect Effect via Pity | −.001 (.012) | [-0.027, 0.027] | .002 (.007) | [-0.011, 0.017] |
| D.A. Indirect Effect via Empathy | **.055 (.020)** | **[0.018, 0.099]** | **.013 (.006)** | **[0.004, 0.027]** |
| G.E. Indirect Effect via Outrage | .024 (.019) | [-0.012, 0.064] | .008 (.006) | [-0.004, 0.022] |
| G.E. Indirect Effect via Hope | **.034 (.016)** | **[0.010, 0.075]** | **.010 (.005)** | **[0.003, 0.023]** |
| G.E. Indirect Effect via Pity | −.000 (.000) | [-0.005, 0.004] | −.000 (.000) | [-0.002, 0.003] |
| G.E. Indirect Effect via Empathy | **.051 (.020)** | **[0.013, 0.095]** | **.012 (.006)** | **[0.004, 0.026]** |
| S.C. Indirect Effect via Outrage | −.018 (.019) | [-0.057, 0.018] | −.006 (.006) | [-0.019, 0.006] |
| S.C. Indirect Effect via Hope | −.006 (.015) | [-0.036, 0.024] | −.002 (.004) | [-0.011, 0.006] |
| S.C. Indirect Effect via Pity | −.001 (.001) | [-0.006, 0.004] | .000 (.000) | [-0.001, 0.005] |
| S.C. Indirect Effect via Empathy | −.010 (.019) | [-0.048, 0.028] | −.003 (.005) | [-0.013, 0.006] |
| Total D.A. Indirect Effect | **.166 (.066)** | **[0.037, 0.294]** | .010 (.031) | [-0.052, 0.072] |
| Total G.E. Indirect Effect | .082 (.066) | [-0.046, 0.214] | .012 (.031) | [-0.048, 0.073] |
| Total S.C. Indirect Effect | −.032 (.066) | [-0.160, 0.099] | −.023 (.031) | [-0.084, 0.037] |

Statistically significant coefficients are shown in bold. D.A. = Discrimination awareness; S.C. = Social class, G.E. = Group efficacy.

towards Roma were found (contrary to Hypotheses 4 and 6). In line with Hypotheses 2 and 3, mediation analyses showed that discrimination awareness was related to greater intentions and participation in solidarity-based actions through outrage towards the situation of Roma and empathy towards Roma people [36,37]. Finally, Hypothesis 5 was confirmed, where group efficacy of non-Roma as allies was related greater solidarity-based actions through hope towards the situation of Roma and empathy towards Roma people [46–48].

## General discussion

This experimental research analyzes how the social class and discrimination awareness towards Roma people awareness influences prejudice towards Roma as well as non-Roma participation in solidarity actions supporting Roma people (Study 1). In addition, we examine the effects of group efficacy on the intentions and participation of the non-Roma majority in solidarity actions in favor of Roma rights (Study 2). Consistent with previous research (e.g., [25,30,46,49]), our results indicate that social class has a significant effect on prejudice towards Roma. Furthermore, the findings reveal that the non-Roma as allies' group efficacy is associated with solidarity-based actions primarily indirectly via hope and empathy (and correlationally, in Study 1) more than directly. The direct effects of this mobilizing variable are small or inconsistent, suggesting that their main role is to create psychological conditions under which mobilizing emotions can emerge (outrage in Study 1; outrage and empathy in Study 2), rather than to directly increase solidarity-based actions by themselves. Future research with a more extend and complex manipulation of group efficacy in line with Hamann et al. [39] suggestions (including social norms, agency for achieving goals more directly, etc.) could contribute to demonstrate if and with which conceptualization group efficacy can directly mobilize allies.

## Effects of social class on prejudice towards Roma

In both studies results indicated that social class attributed to a Roma person had a main effect on prejudice. This highlights the fact that the social class to which a Roma person is attributed determines the attitudes that receives. In line with Urbiola et al. [16], we observe that social class plays a fundamental role in the prejudice against Roma people, where the Roma person of low social class is the profile to which the less positive characteristics – less moral, social or competent – were attributed, arouses more negative and fewer positive emotions, and would be less recommended for a job. Our results emphasize the need to consider intersectionality [20], pointing out that social class plays a major role in the prejudice suffered by ethnic minorities (e.g., [23,25,49]), specifically in the context of the Roma ethnicity (e.g., [16]).

## Effects on solidarity-based actions: the role of discrimination awareness, group efficacy, and emotions

Across studies, the direct patterns of the mobilizing variables on solidarity-based actions –intentions and participation– are mixed, but highly consistent in their indirect path via emotions. Discrimination awareness positively predicted solidarity-based actions intentions overall (in Study 2) and indirectly via outrage (in Study 1) in line with our hypotheses [30,57]. Discrimination awareness had a main effect on solidarity-based actions intentions (e.g., [33]) in Study 2 in line with our predictions, however there was only a positive indirect effect via outrage in Study 1.

Interestingly, in Study 1, when outrage was controlled for, a direct *negative* effect of discrimination awareness on decreasing solidarity-based actions was found. This pattern suggests a suppression effect or defensive processing [56,58], possibly reflecting that some audiences may perceive discrimination claims as illegitimate or as unduly blaming the system in the condition of discrimination awareness [59,60]. Thus, Roma target perceived discrimination can be seen as an act of "self-victimization" or complaining, triggering moral disengagement instead of promoting solidarity [59,60]. For some members of the majority group, confronting allegations of discrimination may threaten beliefs relevant to their identity (e.g., that the social system is generally fair and legitimate [61,62] and trigger defensive processing or resistance to the message [63,64]. Under these conditions, solidarity may be directly inhibited unless the same message also evokes mobilizing emotions– in particular, outrage and empathy– that help reframe the information as a moral concern and a call for solidarity-based actions [30,65]. Importantly, this pattern suggest that testimonies can trigger victim-blaming; underscoring the need to design messages that channel awareness into outrage and empathy while minimizing defensiveness.

Conversely, this effect was not replicated in Study 2 so we cannot draw strong conclusions about it. Overall, when considering the mobilizing emotions that transform the awareness of discrimination into a source of motivation to engage in solidarity actions, discrimination awareness primarily increases intentions indirectly, via outrage (and empathy) whereas effects on actual participation were not direct. This is important because we are referring to discrimination awareness as a consequence of denouncing experienced discrimination by a member of the Roma minority herself. That implies that confronting discrimination by minority members has a powerful effect on motivating the advantaged group as potential allies, at least to the extent that it can trigger anger and outrage in them.

Importantly, our studies for non-Roma allies' group efficacy show a similar pattern of limited direct but meaningful indirect effect on solidarity-based actions. We find correlational evidence in Study 1 indicating that believing that non-Roma as allies was positively related to actively participating in pro-Roma solidarity actions [30,43,53,57]. We did not confirm this effect when group efficacy was experimentally manipulated in Study 2. Regarding the absence of a direct effect of group efficacy and the participation in solidarity actions (in line with previous research as Hamann et al. [39] and Urbiola et al. [46] in some studies), our results however showed consistently that the manipulation of group efficacy worked in base of the manipulation check and operated indirectly via the targeted emotions (hope and empathy), which then predicted intentions and participation (petition signing). This path pattern is also fully consistent with previous literature on collective action, according to which beliefs about efficacy mobilize action primarily by increasing approach-oriented affect (e.g., hope) and empathic concern, rather than exerting an independent main effect [30,47,48,66–68]. However, we found two indirect effects from group efficacy through positive emotions (hope and empathy), to solidarity-based actions

intentions and participation. This suggests that higher efficacy of non-Roma allies can enhance intentions to engage in solidarity-based actions when it is accompanied by emotional engagement, like empathy [36] and hope [47,48]. The emotion of hope has been considered as a moderator and a mediator of group efficacy. The evidence for the role of hope in solidarity-based actions is partial and mixed (e.g., [48]). More evidence is needed to elucidate the role of hope in motivating solidarity-based actions, and to explain its connection to group efficacy. Notably, pity did not mediate this relationship on any of our studies, indicating that is not a motivator for solidarity-based actions participation in Roma context in Spain (contrary to what Lantos et al. [18] found in Hungary). These results highlight that emotions are relevant as underlying psychological factors explaining social mobilization [37,69]. At the same time, the weak, null or negative (for discrimination awareness) direct paths for group efficacy and discrimination awareness underscore a crucial challenge for mobilizing the non-Roma majority, where informing people about discrimination or emphasizing that they can be effective allies could not be enough, at least in the ways that we are testing it and in this intergroup context. Mobilization can require messages that activate the concrete emotional responses while minimizing defensiveness and threat to system justification beliefs.

This research also has limitations, since it only focuses on non-Roma participants in Spain, so we can find members of other minorities (e.g., migrants) in the samples, because we only asked whether or not the participant was Roma, but not whether the participant belonged to another ethnic or national minority. This could be influencing the responses of participation in pro-Roma solidarity-based actions.

Another limitation can be the manipulation of group efficacy of non-Roma people as allies, since group efficacy may not be having direct effects because of the way it is experimentally manipulated. Previous literature has highlighted inconsistent effects in its manipulation (e.g., [39]).

Finally, audiovisual manipulations (accent, clothing, style) could activate or reinforce stereotypes, essentialize categories, and generate discomfort or be interpreted as validation of biases. To mitigate this, at the end of the questionnaire, participants were informed in detail that the person in the video was an actress and that all the contextual information she provided was fictional, except for the historical data and statistics on discrimination against Roma people, which were accurate. Information about the cover story and the real purpose of the study was also provided.

Despite these limitations, the present research contributes to the existing literature in several aspects. To our knowledge, there are no experimental studies examining the effects of social class, discrimination awareness, and group efficacy from the majority group perspective on the prediction of prejudice and pro-Roma solidarity-based actions, considering emotions that mediate the latter. This research manipulated group efficacy from non-Roma majority perspective that has been underexplored in literature. In addition, these solidarity-based actions are measured in two ways: intentions to participate in solidarity-based actions and a measure of actual participation, by signing or not signing a petition in favor of Roma rights. The present research also adopts a novel methodological approach by utilizing video-based manipulations, enhancing the ecological validity of our experimental design while providing valuable insights for future practical interventions. Finally, we can observe how the most disadvantaged groups often experience unique forms of discrimination depending on the social categories of belonging – such as ethnicity, gender, sexual orientation, or social class–, for example, being a Roma from a low social class. This research reinforces the impact of social class on prejudice towards ethnic minorities, specially towards Roma, found by previous research [15,16,70], contributing to understand social inequalities faced by minorities. Our results shows that the prejudice and discrimination that Roma people experience daily are strongly influenced by their perceived social class, commonly attributed to a lower one and reinforce the need to include the attributed social class of ethnic groups as an important factor in the prejudice they face.

Additionally, this research shows the importance of discrimination awareness and group efficacy as two avenues of social mobilization of non-Roma allies for Roma rights through the emotions of outrage and hope towards the situation of Roma people and empathy towards Roma people.

One of the major contributions of this research has been to demonstrate a way to increase the intention as well as participation (e.g., by signing the petition in favor of Roma rights) in solidarity-based actions towards Roma. It should be

considered that there is a low participation in general in the context of activism in favor of Roma rights [10]. In addition, when measuring intentions vs. participation, we must keep in mind that expressing the intention to participate is easier and does not imply an immediate cost, and that the intentions to participate in solidarity-based actions does not always translate into actual behavior [71] although this research also founds effects in the direct participation. It is worth highlighting that, in the context of pro-Roma actions, even a typically low level of mobilization –whether in tendencies or behaviors– represents a significant step forward.

This research with innovative experimental methods provides fundamental keys to mobilizing the non-Roma population to participate as allies in support of Roma rights, as well as when it comes to carrying out interventions and programs by the administrations, encouraging specific emotional approaches, as well as highlighting the discrimination suffered by the Roma population and how non-Roma people can help in the search for social equity.

## Author contributions

**Conceptualization:** Cristina Carmona-López, Ana Urbiola, Marisol Navas.

**Data curation:** Cristina Carmona-López.

**Formal analysis:** Cristina Carmona-López, Ana Urbiola, Soledad de Lemus.

**Funding acquisition:** Ana Urbiola.

**Investigation:** Cristina Carmona-López, Ana Urbiola, Marisol Navas.

**Methodology:** Cristina Carmona-López, Ana Urbiola, Marisol Navas, Soledad de Lemus.

**Project administration:** Ana Urbiola, Soledad de Lemus.

**Resources:** Ana Urbiola, Marisol Navas, Soledad de Lemus.

**Supervision:** Ana Urbiola, Marisol Navas.

**Validation:** Cristina Carmona-López, Ana Urbiola.

**Visualization:** Cristina Carmona-López, Ana Urbiola, Marisol Navas, Soledad de Lemus.

**Writing – original draft:** Cristina Carmona-López, Ana Urbiola, Marisol Navas.

**Writing – review & editing:** Cristina Carmona-López, Ana Urbiola, Marisol Navas, Soledad de Lemus.

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
