## [Decision Letter · Decision Letter 0]

28 Oct 2025

Dear Dr. Carmona-Lopez,

Thank you for submitting your manuscript to PLOS ONE. After careful consideration, we feel that it has merit but does not fully meet PLOS ONE’s publication criteria as it currently stands. Therefore, we invite you to submit a revised version of the manuscript that addresses the points raised during the review process.

The manuscript addresses an important and timely topic with a rigorous and innovative experimental design. The intersectional approach (ethnicity × social class) and focus on emotional mediators meaningfully advance research on anti-Gypsism.  Based on the reviewers' comments, some minor changes would be needed to improve its overall quality.

The introduction should better situate the findings within broader European or international contexts, acknowledging cross-country variations in Roma histories and prejudice. The inconsistent direct effects of the discrimination awareness and group efficacy manipulations warrant fuller discussion—interpreting these null or negative results as informative about the challenges of mobilizing non-Roma solidarity. Clarifying the mediation model specifications will also strengthen transparency.

Finally, ethical and methodological issues related to the visual stimuli (accent, clothing, style) and potential reinforcement of stereotypes should be more explicitly addressed. Expanding on possible “backlash” or “self-victimization” mechanisms using relevant theoretical frameworks (e.g., system justification) would enrich interpretation.

We look forward to receiving your revised manuscript.

Kind regards,

Annalisa Rosso

Academic Editor

PLOS ONE

Journal Requirements:

2. Please describe in your methods section how capacity to provide consent was determined for the participants in this study. Please also state whether your ethics committee or IRB approved this consent procedure. If you did not assess capacity to consent please briefly outline why this was not necessary in this case.

Reviewers' comments:

Reviewer's Responses to Questions

**Comments to the Author**

1. Is the manuscript technically sound, and do the data support the conclusions?

Reviewer #1: Yes

Reviewer #2: Yes

2. Has the statistical analysis been performed appropriately and rigorously?

Reviewer #1: Yes

Reviewer #2: Yes

3. Have the authors made all data underlying the findings in their manuscript fully available?

Reviewer #1: Yes

Reviewer #2: Yes

4. Is the manuscript presented in an intelligible fashion and written in standard English?

Reviewer #1: Yes

Reviewer #2: Yes

Reviewer #1: The article addresses a crucial topic of current social and scientific relevance. The experimental approach and the adoption of an intersectional perspective (ethnicity x social class) fill a gap highlighted in the literature and represent a significant advance in the study of anti-Gypsism.

The research involves exclusively non-Roma samples in Spain. The literature highlights how attitudes and dynamics of prejudice and solidarity can vary significantly across countries, regulatory contexts, and national histories. In the introduction, it would be helpful to include a reference to these possible differences on an international (or at least European) basis. The histories of Roma differ significantly from country to country, and it would be helpful to underline this aspect.

The methodology is very well structured and rigorous; however, it should be remembered that the manipulation of "group efficacy" in Study 2 did not produce direct effects, and an additional manipulation check would be helpful to clarify the phenomenon better. Despite the indirect analyses, the lack of a main effect suggests validity issues in the experimental manipulation. Furthermore, this undermines the reliability of the conclusions on the group efficacy variable and limits the strength of the findings on emotional mediating effects. This point would be better clarified.

Visual methodologies are interesting, but it is helpful to attach (or provide in the data repository) an exact transcription of the materials. The use of videos that manipulate accent, clothing, and speaking style risks reinforcing stereotypes rather than reducing them. A more in-depth analysis of experimental manipulations' ethical consequences and unintended side effects would be desirable.

Beware of exposing the results, which suggest that discrimination can, in some cases, elicit adverse reactions or rejection (self-victimising effect against the Roma protagonist), to "victim blaming." This point, only partially discussed by the authors, should be explored more critically with respect to the risks of awareness campaigns based on storytelling or direct testimonies.

The conclusions are also reasonable. The bibliography is exhaustive.

Reviewer #2: Thank you for the opportunity to review this manuscript. This research addresses a timely and important social issue using a methodologically robust experimental approach. The large samples and the examination of both behavioral intentions and actual participation are significant strengths. The findings regarding the role of attributed social class in prejudice are clear, robust, and constitute a major contribution.

While I am overall very positive about the manuscript, I have some comments and suggestions that I believe will help you strengthen the presentation and interpretation of your complex results, ultimately enhancing the manuscript's impact.

Interpretation of mixed findings for mobilizing variables: the core experimental findings for discrimination awareness and group efficacy are nuanced. While the mediation pathways (via outrage, empathy, and hope) are well-supported, the direct effects of the experimental manipulations are either non-significant (group efficacy in Study 2; discrimination awareness on participation) or negative (discrimination awareness on intentions in Study 1). The discussion would be significantly strengthened by a more direct and detailed confrontation of these null and negative direct effects. What do these inconsistent direct effects tell us about the challenges of mobilizing the non-Roma majority? I recommend refining the conclusions to emphasize that the primary mechanism of action for these variables is indirect, through the specific emotional pathways you identify. A more balanced discussion that integrates both the positive indirect and the null/negative direct effects will provide a more compelling and accurate narrative.

Theoretical implications of the suppression effect: the potential suppression effect in Study 1 is a fascinating finding. The brief mention of "self-victimization" is a plausible starting point, but this section should be expanded. Please consider integrating theories of backlash, system justification, or perceived illegitimacy of claims to provide a deeper theoretical explanation for why simply being made aware of discrimination might, for some, directly inhibit solidarity.

Clarity on statistical model specification: for full rigor and reproducibility, please clarify the exact specification of the mediation models. The captions for Figures 2 and 4 state that analyses were performed "including Social Class Condition" but that it was omitted from the figures. To ensure readers can fully interpret your results, please state explicitly in the method or results section whether the coefficients for discrimination awareness and group efficacy in the mediation models (e.g., in Tables 2 and 4) were derived from models that controlled for the other experimental factors (social class, and in Study 2, the other main factors) as covariates in all paths of the model.

**Do you want your identity to be public for this peer review?** For information about this choice, including consent withdrawal, please see our Privacy Policy

Reviewer #1: **Yes:** prof. Alessandra Sannella, University of Cassino and Southern LazioEUt+ – EUTINN, European University of Technology, European Union

Reviewer #2: No

---

## [Author Response · Author response to Decision Letter 1]

1 Dec 2025

Please refer to the document entitled ‘Response for the reviewers’ for a point-by-point response to the comments and suggestions you made during the review process. Thank you very much.

---

## [Editor Report · Decision Letter 1]

30 Dec 2025

Unpacking antigypsism and support for solidarity-based actions: implications of social class, discrimination awareness, group efficacy and emotions

PONE-D-25-25661R1

Dear Dr. Cristina Carmona-Lopez,

We’re pleased to inform you that your manuscript has been judged scientifically suitable for publication and will be formally accepted for publication once it meets all outstanding technical requirements.

Kind regards,

Annalisa Rosso

Academic Editor

PLOS One

Additional Editor Comments (optional):

The authors have adequately addressed the main comments raised by the reviewers, and the paper can now be considered suitable for publication.
---

## [Editor Report · Acceptance letter]

PONE-D-25-25661R1

PLOS One

Dear Dr. Carmona-López,

I'm pleased to inform you that your manuscript has been deemed suitable for publication in PLOS One. Congratulations! Your manuscript is now being handed over to our production team.

Kind regards,

on behalf of

Dr. Annalisa Rosso

Academic Editor

PLOS One